# Chemical Composition of Turmeric (*Curcuma longa* L.) Ethanol Extract and Its Antimicrobial Activities and Free Radical Scavenging Capacities

**DOI:** 10.3390/foods13101550

**Published:** 2024-05-16

**Authors:** Huan Wu, Zhihao Liu, Yaqiong Zhang, Boyan Gao, Yanfang Li, Xiaohua He, Jianghao Sun, Uyory Choe, Pei Chen, Ryan A. Blaustein, Liangli Yu

**Affiliations:** 1Key Laboratory of Xin’an Medicine, Ministry of Education, Anhui University of Chinese Medicine, Hefei 230038, China; 2Department of Nutrition and Food Science, University of Maryland, College Park, MD 20742, USA; zhihao.liu@usda.gov (Z.L.); yanfang.li@usda.gov (Y.L.); ericchoe88@gmail.com (U.C.); rblauste@umd.edu (R.A.B.); lyu5@umd.edu (L.Y.); 3Methods and Application of Food Composition Laboratory, Beltsville Human Nutrition Research Center, Agricultural Research Service, United States Department of Agriculture, Beltsville, MD 20705, USA; jianghao.sun@usda.gov (J.S.); pei.chen@usda.gov (P.C.); 4Institute of Food and Nutraceutical Science, School of Agriculture and Biology, Shanghai Jiao Tong University, Shanghai 200240, China; yqzhang2006@sjtu.edu.cn (Y.Z.); gaoboyan@sjtu.edu.cn (B.G.); 5Western Regional Research Center, Agricultural Research Service, United States Department of Agriculture, Albany, CA 94710, USA; xiaohua.he@usda.gov

**Keywords:** turmeric, UHPLC-MS/MS, antimicrobial activity, antioxidant, curcumin, demethoxycurcumin, bisdemethoxycurcumin

## Abstract

Turmeric (*Curcuma longa* L.) is a perennial tuberous plant from the genus *Curcuma* (Zingiberaceae) and has been widely used in foods for thousands of years. The present study examined the ethanol extract of turmeric for its chemical composition, antimicrobial activity, and free radical scavenging properties. UHPLC-MS/MS analysis tentatively identified eight compounds in the turmeric extract. Potential antimicrobial effects of 0.1, 1.0, and 10 mg turmeric equivalents (TE)/mL were evaluated in vitro against a variety of Gram-negative bacteria (i.e., *Escherichia coli*, *Klebsiella pneumoniae*, and *Pseudomonas* sp.) and Gram-positive bacteria (i.e., *Enterococcus faecalis*, *Listeria innocua*, and *Staphylococcus aureus*). Concentrations of 0.1 and 1.0 mg TE/mL inhibited the growth of *S. aureus* and significantly suppressed that of *Pseudomonas* sp., *E. faecalis*, and *L. innocua*. The growth of all strains, including *E. coli*, was inhibited by 10 mg TE/mL. Moreover, free radical scavenging capacities were determined using HO^●^, ABTS^●+^, and DPPH^●^ (HOSC, ABTS, and RDSC, respectively) radicals. The turmeric ethanol extract had a TPC value of 27.12 mg GAE/g, together with HOSC, RDSC, and ABTS values of 1524.59, 56.38, and 1.70 μmol TE/g, respectively. Our results suggest that turmeric extract has potential applications for use in functional foods to reduce microbial burdens and oxidative stress-related health problems.

## 1. Introduction

Spices have been an integral part of the human diet for thousands of years. More recently, the growing interest in the relationship between diet and health has elevated the importance of using spices in the food arena and understanding the mechanisms behind their beneficial properties [1]. Turmeric (*Curcuma longa* L.) is a perennial tuberous plant from the genus *Curcuma* (Zingiberaceae) and is commonly distributed in East, South, and Southeast Asia [2]. The rhizomes of turmeric are well-developed, clumped, branching, elliptic, or cylindrical and are widely used in many foods, such as canned beverages, baked products, dairy products, ice cream, yogurt, yellow cakes, milk, orange juice, biscuits, popcorn, cereals, sauces, and others, in the form of powder for over 4000 years [3,4]. The use of turmeric in foods adds a distinctive flavor compared to other spices. Previous chemical analysis of turmeric has shown that it mainly contains terpenoids, curcuminoids, and other phenolic compounds [5]. Among these components, curcuminoids were the major bioactive constituents and recognized by the U.S. Food and Drug Administration as a safe tolerable class of components, at doses up to 8 g per day [6]. Additionally, the main component of turmeric, curcumin, has an orange-yellow color and is an important food colorant. Today, with an increasing interest in applications for food and beverages, as well as nutraceuticals, the global curcumin market is projected to grow from USD 85.77 million in 2023 to USD 165.10 million by 2030 [7].

In addition to enhancing flavor and color in food products, turmeric has applications to support human health, such as through anti-inflammatory, anti-cancer, and antimicrobial effects [8]. For example, ethnopharmacological studies have reported turmeric use as a medical herb for knee osteoarthritis [9], radiodermatitis [10], and cancers [11]. Curcumin has notable antibacterial properties, as it has been shown to inhibit the growth of *Escherichia coli*, *Klebsiella pneumoniae*, and *Staphylococcus aureus*, as well as suppress the formation of mixed-community biofilms and aid in the body’s ability to clear bacteria [12,13]. Curcumin has also been reported to support immunologic response through reducing inflammation, limiting the overexpression of cytokines, and improving the removal of reactive oxygen species, which may lower risks for developing chronic disease [14]. Nevertheless, turmeric, rather than pure curcumin, is typically used as a food ingredient. Characterizing the chemical components and bioactivities of turmeric could provide a foundation for its improved use in food system safety and health-promoting foods and may add profitability to the turmeric industry.

This research aims to understand the antibacterial and radical scavenging activities of turmeric and the chemical components that may contribute to these properties. The antibacterial effects of turmeric extract were evaluated against several Gram-positive and Gram-negative bacteria strains. The radical scavenging properties were examined against ABTS^●+^, DPPH^●^, and HO^●^ radicals according to our published laboratory protocols. In addition, the total phenolic content (TPC) was determined for the extracts as they are potential contributors to overall bioactivity. The results from this study have important implications for the potential utilization of turmeric as an active ingredient in foods to enhance food safety and human health.

## 2. Materials and Methods

### 2.1. Materials

Turmeric (*Curcuma longa* L.) root was gifted from Frontier Co-op (Norway, IA). DPPH^●^, ABTS, Folin–Ciocalteu’s phenol reagent (F9252), gallic acid (G7384), fluorescein (FL), (±)-6-hydroxy-2,5,7,8-tetramethylchromane-2-carboxylic acid (Trolox), ferric chloride (FeCl_3_), and hydrogen peroxide (H_2_O_2_) were purchased from Sigma-Aldrich Corporation (St. Louis, MO, USA). Acetonitrile (LC-MS grade) and formic acid (HPLC grade) were purchased from Merck (Darmstadt, Germany). Other chemicals used in this study were purchased from Fisher Scientific (Hampton, NH, USA), at analytical grade. Bacteria used in the antimicrobial assays included *E. coli* K-12 and strains of *Enterococcus faecalis*, *K. pneumoniae*, *Listeria innocua*, *Pseudomonas* sp., and *S. aureus*, maintained in glycerol stocks at the University of Maryland, College Park.

### 2.2. Sample Preparation

Turmeric root was grounded to powder using a Micromil Grinder (Wayne, NJ, USA). The grinding process was repeated until all the turmeric root powder passed No. 40 mesh (0.420 mm). Next, 1 g of turmeric powder was accurately weighed and mixed with 10 mL of ethanol for 24 h at ambient temperature. After 24 h, a centrifuge (3500 rpm) was used to obtain turmeric extract having a concentration of 100 mg dry turmeric equivalents (TE)/mL and stored at −20 °C.

### 2.3. Chemical Compositions of Turmeric (Curcuma longa L.) Ethanol Extract

The chemical compositions of turmeric ethanol extract were analyzed with a Vanquish UHPLC- Orbitrap Fusion ID-X Tribrid mass spectrometer. The sample was separated in an Agilent Eclipse Plus-C_18_ column (150 mm × 2.1 mm i.d., 1.8 µm). Mobile phase consisted of 0.1% formic acid in acetonitrile (*v*/*v*) (A) and 0.1% formic acid in water (*v*/*v*) (B). The gradient elution procedure is as follows: 0 min (2%A), 15 min (10%A), 35 min (40%A), 55 min (95%A), and 60 min (95%A), then re-equilibration with 2%A for 10 min. Conditions were as follows: Injection volume, 1 µL. Flow rate, 0.3 mL/min. Mass Spectrometry conditions: Resolution, 60,000. Scan range, *m*/*z* 120 to 1200. Ion transfer tube temperature, 300 °C. Ion source temperature, 275 °C. The capillary voltages were 3.9 kV (positive ion mode) and 2.5 kV (negative ion mode). Peaks were tentatively identified by their precision mass of excimer ions, MS/MS data and fragmentation patterns, and data from the literature.

### 2.4. Antimicrobial Activities

Antimicrobial effects of turmeric were evaluated against three Gram-negative (*E. coli*, *K. pneumoniae*, *Pseudomonas* sp.) and three Gram-positive (*E. faecalis*, *L. innocua*, *S. aureus*) bacteria. Isolates from pure cultures of the respective strains were individually grown in 500 μL trypticase soy broth (TSB) incubated at 37 °C for 24 h. The pre-grown cultures were vortexed, and 10 μL was transfer-inoculated into 180 μL TSB and that which was augmented with varying concentrations of turmeric ethanol extract dissolved in DMSO (i.e., 0.1, 1, and 10 mg turmeric equivalents mL^−1^) in 96-well plates. Since DMSO can have antimicrobial effects at high concentrations [15], TSB containing 0, 0.1, 1, and 10% DMSO with no turmeric was included as assay controls. All plates were incubated at 37 °C shaking at 180 rpm for 24 h. Optical density absorbance at 600 nm (OD_600_) was measured with a Multiscan FC plate reader (Thermo Scientific, Waltham, MA) at 0, 3, 6, 12, and 24 h. OD_600_ values reflecting bacterial growth were normalized against negative control blanks that contained no bacteria. All assays were performed with three biological replicate plates, in which technical replicates for each treatment were averaged.

### 2.5. Total Phenolic Content (TPC)

The TPC of turmeric ethanol extract was measured using a laboratory protocol, as previously described [16]. Briefly, 3 mL of water was mixed with 50 μL of solvent (blank), standard, or sample. To the mixture, 250 μL of Folin–Ciocalteu’s phenol reagent was added. After the vortex, 750 μL of 20% (*w*/*v*) Na_2_CO_3_ was added to initiate the reaction, and the reaction mixture was maintained in ambient temperature under dark for 2 h. After the incubation, the absorbance at 765 nm was measured. The results were calculated based on the standard curve generated using gallic acid and reported in milligrams of gallic acid equivalents (GAE) per gram of dry turmeric sample (mg GAE/g turmeric).

### 2.6. Free Radical Scavenging Capacities

The 2,2′-azino-bis (3-ethylbenzothiazoline-6-sulfonic acid) radical (ABTS^●+^) values were detected following a laboratory protocol [16]. Briefly, 160 μL of solvent/standards (5 to 300 µmol/L Trolox solution)/clove extracts was mixed with 2 mL of ABTS^●+^ working solution by vortexing for 30 s. Following a 60 s reaction, the absorbance was detected at 734 nm with a Genesys 20 visible spectrophotometer (Thermo Fisher Scientific, Norristown, PA, USA). The relative 2,2-diphenyl-1-picrylhydrazyl (DPPH) radical scavenging capacity (RDSC) values were detected based on previous studies [17,18]. Briefly, 100 μL of solvent, standards (7 to 36 µmol/L Trolox solution), or sample was mixed with 100 μL of freshly prepared 0.2 mM DPPH solution in the wells of a 96-well plate. Detection wavelength was 515 nm, and the samples were detected every minute for 90 min using a Tecan M200 Pro microplate reader (Tecan Group Ltd., Mannedorf, Switzerland). The hydroxyl radical scavenging capacity (HOSC) values were detected, as previously described [19]. Briefly, 170 μL of freshly prepared fluorescein working solution (92.8 nM) was mixed with 30 μL of either solvent, standard, or sample in the wells of a 96-well plate. Then, 40 μL of freshly prepared H_2_O_2_ working solution (199 mM) and 60 μL of FeCl_3_ (3.43 mM) were added to start the reaction. The fluorescence intensities were recorded (excitation: 485 nm, emission: 535 nm) every five minutes for 8 h with a Tecan M200 Pro microplate reader (Tecan Group Ltd., Mannedorf, Switzerland). In all three assays, Trolox was used as a standard, and results were expressed as micromoles of Trolox equivalents (TE) per gram of dry turmeric sample (μmol TE/g turmeric).

### 2.7. Statistical Analysis

To characterize the antimicrobial effects of turmeric, area under the curve (AUC) for logistic bacterial growth (OD_600_) was approximated using Growthcurver [20]. ANOVA with Tukey’s post hoc test was applied to evaluate differences in AUC, as well as in OD_600_ for each time point (i.e., 3, 6, 12, 24 h), based on the augmented concentrations of turmeric and/or DMSO in the growth media. The Student’s *t*-test was further used to determine differences in AUC between the turmeric treatments and respective DMSO controls (e.g., compare 10 mg ml^−1^ turmeric dissolved in DMSO in TSB to 10% DMSO in TSB, as both contained equal volumes DMSO in the bacterial growth media). These statistical analyses were performed in R.v.4.3.1. For radical scavenging properties and TPC, *t*-test was performed using the GraphPad Prism 9 software.

## 3. Results and Discussion

### 3.1. Chemical Compositions of Turmeric Ethanol Extract

Eight compounds (Table 1) were tentatively identified in turmeric ethanol extract using tandem mass spectrometry and literature data. All eight compounds were phenolic compounds. Compound **1** and compound **5** were identified as coumaric acid [21] and calebin A, respectively. Both coumaric acid and calebin A were reported in turmeric ethanol extract for the first time. Compounds **2**–**4** were keto forms of bisdemethoxycurcumin, demethoxycurcumin, and curcumin, while compounds **6**–**8** were enol forms of bisdemethoxycurcumin, demethoxycurcumin, and curcumin [22,23]. These results agreed with the previous reports that curcuminoids are the major bioactive compounds found in turmeric root [24]. In addition, our study showed that the ion intensities of the enol form of curcuminoids were higher than the keto form of curcuminoids. Compound **6** (bisdemethoxycurcumin) had the highest intensity, followed by compound **7** (demethoxycurcumin) and compound **8** (curcumin) in the total ion chromatogram (TIC) (Figure 1).

Compound **5** had *m*/*z* values of 383.1111 and 385.1257 in negative and positive ion mode, respectively (Table 1). These values coincided with the molecular formula of C_21_H_20_O_7_ and it was speculated to be calebin A by searching in SciFinder. For its product ions (MS^2^) in positive mode (Table 1), the *m*/*z* values were 261.0737 [M+H–C_7_H_9_O_2_]^+^ and 177.0531 [M+H–C_11_H_12_O_4_]^+^, which are the product ions of the loss of 2-methoxyphenol and methyl 3-(4-hydroxy-3-methoxyphenyl)acrylate from calebin A, respectively.

Compound **6** showed the highest intensity, in which bisdemethoxycurcumin had *m*/*z* values of 307.0954 and 309.1100 in negative ([M−H]^−^) and positive ([M+H]^+^) modes, respectively. These values coincided with the molecular formula of C_19_H_16_O_4_ (Table 1). For its product ions (MS^2^) in negative mode, *m*/*z* values of 187.0383, 143.0488, and 119.0491 were detected. It has been reported that *m*/*z* values of 187.0383 and 119.0491 were obtained by cleavage of the bond between C_2_ and C_3_ of bisdemethoxycurcumin, whereas the *m*/*z* value of 143.0488 was obtained by the loss of CO_2_ from the *m*/*z* 187.0383 fragment [22] (Figure 2). For bisdemethoxycurcumin’s MS^2^ in positive ion mode, two fragments, *m*/*z* 225.0892 and *m*/*z* 147.0428, were detected (Table 1). It has been reported that the fragment of *m*/*z* 225.0892 is generated by the loss of a 1-hydroxy-3-ketocyclobutene moiety from its precursor ion (MS), *m*/*z* 309.1100. Contrarily, an *m*/*z* 147.0428 fragment was generated by a C_3_ and C_4_ bond cleavage followed by a neutral loss of a 1-aryl-3-hydroxy-1,3-butadiene moiety [24]. By comparing MS and MS^2^ results with the previously published literature, compound **6** was tentatively identified as bisdemethoxycurcumin.

Two other curcuminoids (compounds **7** and **8**) that showed the second and third highest peak intensity in TIC, corresponding to demethoxycurcumin and curcumin, had *m*/*z* values of 337.1057 and 367.1157 in negative ion mode, respectively (Table 1). These *m*/*z* values are 30 and 60 higher than bisdemethoxycurcumin due to one methoxyl group found in demethoxycurcumin and two methoxyl groups found in curcumin. In the same way, both demethoxycurcumin and curcumin showed product ions having *m*/*z* values of 173 and 217 compared to bisdemethoxycurcumin’s *m*/*z* 143 and 187 fragments due to methoxyl group differences (Table 1). Using the same principle as described in identifying compound **6**, all other compounds found in turmeric ethanol extract were tentatively identified. Intriguingly, three curcuminoids, such as bisdemethoxycurcumin, demethoxycurcumin, and curcumin, were detected twice at different retention times. For instance, bisdemethoxycurcumin was detected at a retention time of 31.44 and 39.08 min, demethoxycurcumin at 32.21 and 39.79 min, and, lastly, curcumin at 33.28 and 40.44 min (Table 1). These phenomena had previously explained that these curcuminoids exist as rapidly interconverting keto-enol tautomers [23].

Previously, Yang et al. extracted turmeric powder using 80% ethanol and found seven compounds, including gallic acid, protocatechuic acid, epicatechin, rutin, curcumin, myricetin, and cinnamic acid [25]. Only one curcuminoid (curcumin) was detected in this study. In the present study, three curcuminoids, including bisdemethoxycurcumin, demethoxycurcumin, and curcumin, were detected. These differences may be partially due to the different extraction solvents, different analytical methods, or the material variations.

### 3.2. Antimicrobial Activities

The antimicrobial effects of turmeric ethanol extract in a range of practical concentrations (0.1, 1, and 10 mg mL^−1^) were evaluated against a variety of Gram-negative and Gram-positive bacteria. The growth of all bacteria was inhibited by turmeric at the highest concentration of 10 mg mL^−1^ (dissolved in DMSO), while the lower concentrations appeared to suppress bacterial growth in a strain-specific manner (Figure 3).

*Pseudonomas* sp. was the only Gram-negative strain in which growth was abrogated at 0.1 and 1 mg mL^−1^ (Figure 3A). At the lowest concentration, OD_600_ for *Pseudonomas* sp. was lower at 3 h compared to that of the strain when grown in TSB without turmeric (*p* < 0.024), demonstrating a subtle yet significant effect on lag-stage time for its growth. Similarly, at 1 mg mL^−1^, OD_600_ for *Pseudomonas* sp. was significantly lower at 3 h (*p* = 0.013), 6 h (*p* = 0.043), and 24 h (*p* = 0.025) compared to the respective control, yielding a significantly lower AUC (*p* = 0.005) (Figure 3B). Thus, while *E. coli* and *K. pneumoniae* appeared to tolerate turmeric at equivalents of 0.1 and 1 mg mL^−1^ (*p* >0.05 for all time points and for AUC), the growth of *Pseudomonas* was slowed by the lowest concentration and significantly suppressed at the medium level tested.

The Gram-positive bacteria had different growth patterns and appeared more sensitive to turmeric (Figure 3A). At 0.1 mg mL^−1^, OD_600_
*E. faecalis* was lower at 6 h (*p* < 0.001), that of *L. innocua* was lower at 12 h (*p* = 0.039), and that of *S. aureus* was lower at 6 h (*p* = 0.032), 12 h (*p* < 0.001), and 24 h (*p* < 0.001). In media containing turmeric at 1 mg mL^−1^, OD_600_ of *E. faecalis* was lower at 6 h (*p* < 0.001) and 12 h (*p* = 0.007), and that of *S. aureus* was lower at 12 h (*p* < 0.001) and 24 h (*p* < 0.001). Accordingly, the AUC values for growth curves for *S. aureus* in media with all concentrations of turmeric tested were significantly suppressed relative to the control (*p* < 0.001 for each concentration) (Figure 3B). Thus, 0.1 to 1 mg mL^−1^ turmeric appeared to have at least some level of antimicrobial activity against all of the Gram-positive bacteria tested, either in the form of complete inhibition (i.e., *S. aureus*) or limiting kinetics for bacterial growth (i.e., *E. faecalis*, *L. innocua*).

The antimicrobial effects of turmeric at the lower concentrations (i.e., 0.1 and 1 mg mL^−1^) were validated with controls in which media augmented with DMSO alone at 0.1% and 1% (i.e., final concentration equivalents to the amount used to dissolve turmeric) had no effects on bacterial OD_600_ at any time point (*p* > 0.05 for all) or on AUC (*p* > 0.05 for all) (Appendix A). We note that 10% DMSO had suppressive, though not entirely inhibitory, effects on bacterial growth (Appendix A). Thus, comparing growth curves with turmeric at 10 mg mL^−1^ was necessary to distinguish the specific antimicrobial properties of the food ingredient. While there were trends for a lower average AUC for all bacteria grown in turmeric at 10 mg mL^−1^ compared to 10% DMSO, the difference in AUC was only significant for *E. coli* (*p* = 0.017) and *S. aureus* (*p* < 0.001) (Figure 4), perhaps reflecting variation in spectrophotometer absorbance values and/or growth at the highest concentration tested. Nevertheless, the antimicrobial effects on AUC for *E. coli*, along with concentration- and time-dependent effects on growth of *Pseudomonas* sp., *E. faecalis*, *L. innocua*, and *S. aureus*, were validated. We note that determining antimicrobial potential against *K. pneumoniae* was limited by the scope of the assay (i.e., the only strain tested in which growth abrogation could not be distinguished from that induced by DMSO). Overall, this study demonstrates that turmeric has antimicrobial activity against a wide spectrum of commonly occurring opportunistic pathogens.

The antimicrobial properties of turmeric have important food safety implications. The *E. coli* K-20 and *L. innocua* strains used in this study are surrogates for predicting the behavior of enteropathogenic *E. coli* and *Listeria monocytogenes* [26,27]. Although generic *E. coli* bacteria are a commonly occurring commensal, pathogenic strains linked to the consumption of contaminated or improperly prepared foods, such as *E. coli* O157:H7, can cause illnesses, such as stomachache, bloody diarrhea, and vomiting. In addition, *L. monocytogenes* and even *L. innocua* can cause symptoms, such as fever, diarrhea, vomiting, nausea, and even death [28]. *Listeria* species and *E. coli* are often linked to the consumption of contaminated leafy vegetable and fresh-cut produce, as well as undercooked meats, among other food products [29]. Our results demonstrate that 10 mg mL^−1^ turmeric was able to significantly inhibit the growth of *E. coli*, and lower concentrations were even suppressive to *L. innocua*. When considering using turmeric in food systems such as turmeric salad dressing, one teaspoon of turmeric powder is often used [30]. The equivalent to 5.7 g of turmeric powder used to prepare the salad dressing, as dissolved in ½ cup of oil and ¼ cup of vinegar, yields a final concentration of approximately 32 mg mL^−1^, which is three-times higher than the highest concentration (10 mg mL^−1^) used in the current study. Thus, in addition to the well-documented beneficial properties of turmeric on consumer health [8], basic use as a food ingredient may have antimicrobial applications to mitigate the risks for foodborne pathogen contamination.

The other strains evaluated in this study were largely members or phylogenetic relatives of ESKAPE pathogens—*Enterococcus faecium*, *S. aureus*, *K. pneumoniae*, *Acinetobacter baumanii*, *Pseudomonas aeruginosa*, and *Enterobacter* species—which, as a group, are the leading cause of nosocomial infections worldwide [31]. *K. pneumoniae* is found in soil, skin, and foods and has been linked to a variety of diseases, including pneumonia, urinary tract infection, diarrhea, meningitis, and sepsis. Although *K. pneumoniae* is recognized as non-food-associated bacteria, a recent infection report indicated that the infection is possibly due to the food working as a transmission vector [32]. While our results were inconclusive as to whether turmeric can inhibit *K. pneumoniae*, which appeared to tolerate low concentrations of the extract, trends for the highest concentration tested, although insignificant, suggest the potential may exist. Moreover, *Pseudomonas* is another important Gram-negative opportunistic pathogen group, in which members such as *P. aeruginosa* can cause pneumonia and sepsis in immunocompromised patients [33]. While *P. aeruginosa* has been rarely associated with foodborne diseases, it is still a common spoilage agent in foods with high water activity and nutrient contents, such as milk, meat, fruits, and vegetables [34]. Given the observed sensitivity of the *Pseudomonas* strain in this study to turmeric, even at relatively low concentrations, applications for the extract as a flavor supplement and preservative agent would be an interesting future direction.

The other bacteria evaluated in this study, *E. faecalis* and *S. aureus*, may have implications for foods with high salt concentrations. *E. faecalis* is widely found in fermented foods such as cured meats and cheeses. Contamination by *E. faecalis* often occurs during food processing [35]. For example, fermented meat products such as salami and Landjager are processed without heat in many cases and have been reported to carry 10^2^ to 10^5^ colony-forming units/g (CFU/g) of *E. faecalis* [36]. Even when heat is applied, *E. faecalis* can survive if the population level is intrinsically high due to its stress tolerance abilities against temperature, pH, and salinity [35]. Similarly, *S. aureus* is a human skin commensal that is widespread in the built environment such as on surfaces in processing facilities [37]. Some strains of *S. aureus* can produce toxins that cause human illness. Because the illness caused by *S. aureus* is typically due to the toxins produced, antibiotics do not work as treatment [37]. A previous study reported that turmeric’s major bioactive component, curcumin ranging from 125 to 250 µg/mL, was able to inhibit the growth of several different strains of *S. aureus* [38]. Likewise, the current study found that all evaluated concentrations (0.1, 1.0, and 10 mg mL^−1^ equivalents) had significant inhibitory effects against *S. aureus*. Thus, the extract may have applications for biocontrol across a wide array of food types, ranging from fresh produce (*E. coli* and *L. innocua*) to high water activity (*Pseudomonas*) and even salt contents (*E. faecalis*, *S. aureus*).

In summary, the turmeric extract exerted antimicrobial activities against a variety of model Gram-negative and Gram-positive bacteria. Although there are limitations for applications of turmeric as functional food ingredients, such as consumer acceptance and solubility (e.g., DMSO was used here), applications for turmeric among other herbs and spices with effective antibacterial activity warrants investigation to improve food safety and further support preservation.

### 3.3. TPC and Free Radical Scavenging Capacities

The phenolic compounds are the most abundant secondary metabolites widely found in the plant kingdom. Because their role is to protect plants from environmental stresses, such as light, extreme temperatures, and pathogen infection, phenolic compounds can be applied as numerous food preservatives. In the present study, the TPC of turmeric ethanol extract was 27.12 mg GAE/g turmeric (Table 2). This value is greater than 6.57 mg GAE/g turmeric reported in the 95% *v*/*v* ethanol extract of turmeric [39]. In another study, Akter and others evaluated six different species and varieties of turmeric using pure methanol as a solvent and found that TPC values were in the range of 37.9–157.4 mg GAE/g turmeric [40]. These differences in results may be related to different extraction solvents and the potential effects of turmeric genotype and growing conditions. 

The evaluation of free radical scavenging capacities is important for both food safety and quality. In addition, radical scavenging components may benefit human health, as redox homeostasis is a basic requirement for performing various normal cellular functions. Free radicals may increase oxidative stress, which may induce the risk of many aging-associated human diseases through the activation of related signaling pathways, immune disorders, DNA mutations, etc. [41]. In foods, free radicals may act as initiators of lipid peroxidation and, consequently, reduce their shelf stability. In the current study, the turmeric extract had ABTS, RDSC, and HOSC values of 1.70, 56.38, and 1524.59 μmol TE/g turmeric, respectively (Table 2). HOSC value was examined for turmeric ethanol extract for the first time. The ABTS and RDSC values are greater than that of 3.25 and 7.23 μmol TE/g turmeric, respectively, reported previously for the turmeric water extract [42]. In addition, the HOSC of turmeric ethanol extract was much lower than that of 2181.08 μmol TE/g for clove ethanol extract [43] but much greater than that of 364.64 μmol TE/g for the honeysuckle ethanol extract [44]. Hitherto, the free radical scavenging capacity of turmeric toward the hydroxyl radical has been attributed to its major bioactive components, curcuminoids [45,46]. For instance, Agnihotri and Mishra assessed the free radical scavenging mechanism of curcumin against the hydroxyl radical. In their study, they found that not only curcumin itself works as an antioxidant for hydroxyl radicals but, also, its degradation products from curcumin including ferulic acid and vanillin can work together as antioxidants [46]. Consequently, bioactive compounds found in turmeric may extend the shelf life of foods since many foods using turmeric are cooked with either oil or fat. In addition, the use of turmeric in preparing foods can also aid in maintaining good health conditions by reducing oxidative stress. Oxidative stress is caused when the body’s redox homeostasis is imbalanced. This can happen through different factors, including smoking, alcohol drinking, and exposure to environmental contaminants, as well as sunlight [47]. Basically, these factors generate excessive reactive oxygen species. Therefore, to prevent oxidative stress, more antioxidants are needed. However, there is a limit to the antioxidants that the body can produce. Considering this, extra sources of antioxidants are necessary, and foods such as turmeric can serve as an extra source.

## 4. Conclusions

Eight phenolic compounds (including coumaric acid, bisdemethoxycurcumin (keto form), demethoxycurcumin (keto form), curcumin (keto form), calebin A, bisdemethoxycurcumin (enol form), demethoxycurcumin (enol form) and curcumin (enol form)) were identified in the ethanol extract of turmeric through UHPLC-MS/MS analysis in this study. Coumeic acid and calebin A were reported in the turmeric ethanol extract for the first time. The ethanol extract of turmeric demonstrated concentration-dependent inhibitory effects against Gram-negative bacteria, including *E. coli* and *Pseudomonas* sp., as well as Gram-positive bacteria, such as *E. faecalis*, *L. innocua*, and *S. aureus*. The antibacterial activity of turmeric ethanol extract against *E. faecalis* and *L. innocua* are reported for the first time in the present study. The ethanol extract of turmeric also contained higher TPC and showed scavenging activities against HO^●^, ABTS^●+^, and DPPH^●^. HO^●^ scavenging capacity was reported for the turmeric extract for the first time. The results suggest that turmeric and its extracts may have applications for use as antibacterial agents in foods to prevent food spoilage and reduce risks for food safety. At the same time, the use of turmeric in home-cooked dishes may provide potential health benefits by quenching excessive free radicals. Exploring how metabolites of turmeric compounds can provide potential health benefits would be an interesting future direction.

## Figures and Tables

**Figure 1 foods-13-01550-f001:**
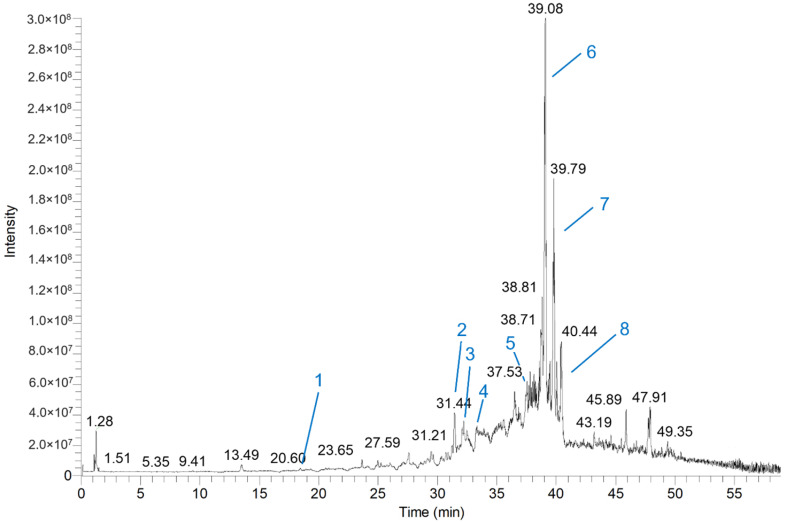
Turmeric ethanol extract’s total ion chromatogram (TIC) operated in the negative ion mode. Numbers (blue) shown on the top of the peak indicate the order of tentatively identified compounds according to their retention time shown in Table 1.

**Figure 2 foods-13-01550-f002:**
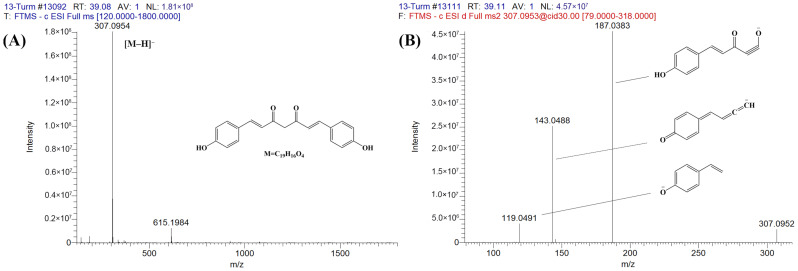
Identification of bisdemethoxycurcumin (C_19_H_16_O_4_). (**A**) Full-scan MS and (**B**) MS^2^ in negative ionization mode.

**Figure 3 foods-13-01550-f003:**
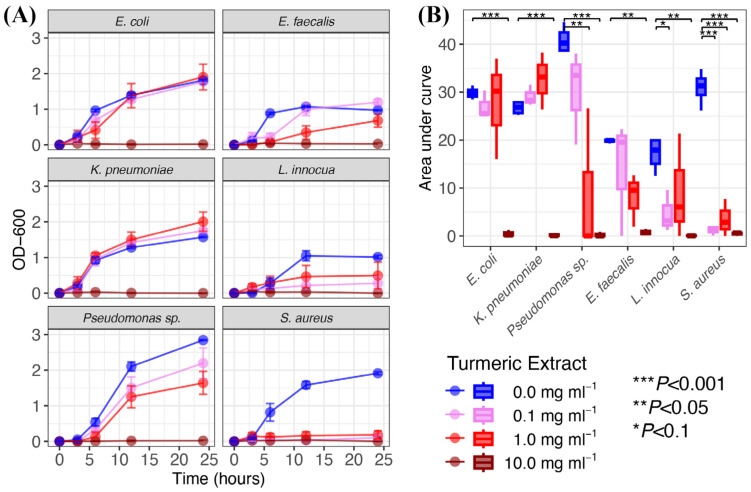
Antimicrobial activity of turmeric ethanol extract dissolved in DMSO. (**A**) Bacterial growth (optical density at 600-nm wavelength; OD-D00) in trypticase soy broth (TSB) supplemented with the dissolved extract. Error bars represent standard error. (**B**) Boxplots for area under the curve. Colors correspond to treatment. The ***, **, and * indicate *p* < 0.001, *p* < 0.05 and *p* < 0.1, respectively.

**Figure 4 foods-13-01550-f004:**
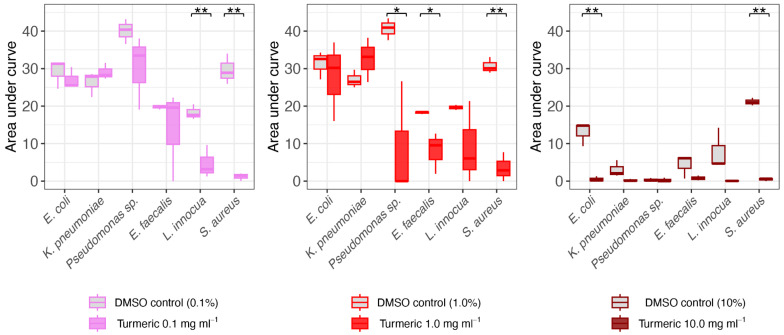
Antimicrobial effects of turmeric ethanol extract dissolved in DMSO (color boxes) relative to DMSO controls at the same concentration (gray boxes). Colors correspond to treatment. The ** and * indicate *p* < 0.05 and *p* < 0.1, respectively.

**Table 1 foods-13-01550-t001:** Characterization of chemical compounds in turmeric (*Curcuma longa* L.) ethanol extract.

Peak No.	*t*_R_ (min)	Negative Ionization Mode			Positive Ionization Mode			Formula	Tentative Identification	Ion Intensity (×10^5^) *	Ref.
		[M−H]^−^	Product Ions	Mass Error (mmu)	[M+H]^+^	Product Ions	Mass Error (mmu)				
1	18.49	163.0386	119.0489	−1.55	ND	ND	ND	C_9_H_8_O_3_	Coumaric acid	12	[21]
2	31.44	307.0952	187.0386 143.0490 119.0492	−2.15	309.1097	225.0894 147.0429	−2.40	C_19_H_16_O_4_	Bisdemethoxycurcumin (keto form)	217	[22]
3	32.21	337.1056	217.0483 173.0588 143.0485	−2.58	339.1205	255.0994 245.0789 239.1071 231.1277 146.3248	−2.22	C_20_H_18_O_5_	Demethoxycurcumin (keto form)	135	[22]
4	33.28	367.1156	217.0484 173.0589 135.0436	−3.08	369.1309	299.1252 285.1097 259.0942 245.0787 175.0737	−2.40	C_21_H_20_O_6_	Curcumin (keto form)	75	[22]
5	37.53	383.1111	217.0487 173.0592 165.0542 158.0358	−2.57	385.1257	261.0737 177.0531	−2.45	C_21_H_20_O_7_	Calebin A	210	
6	39.08	307.0954	187.0383 143.0488 119.0491	−2.15	309.1100	225.0892 147.0428	−2.09	C_19_H_16_O_4_	Bisdemethoxycurcumin (enol form)	1810	[23]
7	39.79	337.1056	217.0488 187.0384 173.0593 143.0489	−2.58	339.1205	255.0994 245.0787 175.0738 147.0426	−2.19	C_20_H_18_O_5_	Demethoxycurcumin (enol form)	1230	[23]
8	40.44	367.1157	217.0486 175.0383 173.0591	−3.05	369.1309	299.1256 285.1100 245.0789 175.0739	−2.41	C_21_H_20_O_6_	Curcumin (enol form)	506	[23]

*t*_R_ represents retention time; ND represents not detected; * generated in negative mode.

**Table 2 foods-13-01550-t002:** Total phenolic content (TPC) and free radical scavenging capacities of turmeric ethanol extract.

TPC (mg GAE/g Turmeric*)	Free Radical Scavenging Capacities (μmol TE/g Turmeric)		
	ABTS	RDSC	HOSC
27.12 ± 0.52	1.70 ± 0.58	56.38 ± 1.18	1524.59 ± 29.89

Turmeric* stands for turmeric ethanol extract. TPC stands for total phenolic content. ABTS, RDSC, and HOSC stand for ABTS^●+^ radical scavenging capacity, relative DPPH scavenging capacity, and hydroxyl radical scavenging capacity, respectively. The final concentrations used for ABTS, RDSC, and HOSC assays were 7.4, 50, and 10 mg dry turmeric equivalents/mL, respectively. TE stands for Trolox equivalents. The results are reported in mean ± standard deviation (n = 3).

## Data Availability

The original contributions presented in the study are included in the article and Appendix A, further inquiries can be directed to the corresponding authors.

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
