# Peer review of "Chemical Composition of Turmeric (Curcuma longa L.) Ethanol Extract and Its Antimicrobial Activities and Free Radical Scavenging Capacities"

_foods, 2024, doi:10.3390/foods13101550_

Round 1

Reviewer 1 Report

Comments and Suggestions for Authors

The article 'Chemical Composition of Turmeric (Curcuma longa L.) Ethanol Extract and Its Antimicrobial Activities and Free Radical Scavenging Capacities' by Wu et al. submitted for review describes the phenolic composition, antioxidant and antimicrobial activities of the ethanolic extract of tumeric.

The article is correctly written, but its main problem is that it does not contain any news. It does not contain any new findings that would advance knowledge in this research area. The purpose of the work is a set of listed methods that will be used to assess the raw material. However, there is no hypothesis and no description of the research problem. Tumeric is a well-studied raw material. The analyses of this raw material carried out in a rather truncated area in this thesis have already been described many times. Furthermore, the citation of works in the methodology is also inadequate. Are the authors of this paper co-authors of the method for evaluating TPC and antioxidant activity (radical scavenging properties against ABTS•+ , DPPH• , and HO• radicals)?  These are well-known methods, developed by others.

Comments on the Quality of English Language

Minor modification is needed

Author Response

Dear reviewer,

Thank you for your comments on our manuscript entitled “Chemical Composition of Turmeric (Curcuma longa L.) Ethanol Extract and Its Antimicrobial Activities and Free Radical Scavenging Capacities” (Manuscript ID: foods-2998792). The comments are all valuable and very helpful for revising and improving our paper. We have revised the manuscript and completed the language editing accordingly / checked the English expression carefully. In the marked-up manuscript, the amendments are highlighted in red. The responses to the reviewers' comments are listed in “foods-2998792-list of itemized responses to reviewers' comments”.

Reviewer 2 Report

Comments and Suggestions for Authors

The manuscript "Chemical Composition of Turmeric (Curcuma longa L.) Ethanol 2 Extract and Its Antimicrobial Activities and Free Radical Scavenging Capacities" by Wu et al, is in a publishable format. I recommend acceptance for publication 

Author Response

Comments and Suggestions for Authors

The manuscript "Chemical Composition of Turmeric (Curcuma longa L.) Ethanol 2 Extract and Its Antimicrobial Activities and Free Radical Scavenging Capacities" by Wu et al, is in a publishable format. I recommend acceptance for publication

Answers: thank you very much.

Reviewer 3 Report

Comments and Suggestions for Authors

The manuscript entitled “Chemical Composition of Turmeric (Curcuma longa L.) Ethanol Extract and Its Antimicrobial Activities and Free Radical Scavenging Capacities” by Wu et al. presents an interesting subject and intends to provide information about the bioactive potential of turmeric extracts for using in functional foods.

The authors evaluated important parameters, which makes this work more robust and present appropriate research design to develop the work. Nevertheless, what is the novelty of the present work? The authors should clearly identified it in the introduction.

The paper is well discussed with comparisons with the reported literature, although the authors should improve grammatically the manuscript and reformulate some sentences to make the idea clearer and succinct. (Examples of sentences that authors should reformulate: Line 244-247, Line 368-369, Line 401-403)

In the Section 2.4., the authors should describe the concentration of the bacteria suspensions and the concentration of the inoculum in the microplate.

Line 418: degradation products

Conclusions: the identified compounds should be described

Based on the mentioned comments, I consider that the authors should revise the manuscript to improve its quality in Foods.

Author Response

(The authors gave the same response as above.)
